# Fault Diagnosis of a Helical Gearbox Based on an Adaptive Empirical Wavelet Transform in Combination with a Spectral Subtraction Method

**Peng Wang and Chang-Myung Lee ***

Department of Mechanical and Automotive Engineering, University of Ulsan, 93 Daehak-ro, Nam-Gu, Ulsan 44610, Korea; wangpeng4468@163.com
* Correspondence: cmlee@ulsan.ac.kr

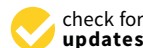

**Featured Application: This work can extract fault features of vibration signals of rotating machinery.**

**Abstract:** Fault characteristic extraction is attracting a great deal of attention from researchers for the fault diagnosis of rotating machinery. Generally, when a gearbox is damaged, accurate identification of the side-band features can be used to detect the condition of the machinery equipment to reduce financial losses. However, the side-band feature of damaged gears that are constantly disturbed by strong jamming is embedded in the background noise. In this paper, a hybrid signal-processing method is proposed based on a spectral subtraction (SS) denoising algorithm combined with an empirical wavelet transform (EWT) to extract the side-band feature of gear faults. Firstly, SS is used to estimate the real-time noise information, which is used to enhance the fault signal of the helical gearbox from a vibration signal with strong noise disturbance. The empirical wavelet transform can extract amplitude-modulated/frequency-modulated (AM-FM) components of a signal using different filter bands that are designed in accordance with the signal properties. The fault signal is obtained by building a flexible gear for a helical gearbox with ADAMS software. The experiment shows the feasibility and availability of the multi-body dynamics model. The spectral subtraction-based adaptive empirical wavelet transform (SS-AEWT) method was applied to estimate the gear side-band feature for different tooth breakages and the strong background noise. The verification results show that the proposed method gives a clearer indication of gear fault characteristics with different tooth breakages and the different signal-noise ratio (SNR) than the conventional EMD and LMD methods. Finally, the fault characteristic frequency of a damaged gear suggests that the proposed SS-AEWT method can accurately and reliably diagnose faults of a gearbox.

**Keywords:** helical gearbox; spectral subtraction; empirical wavelet transform; feature extraction

## 1. Introduction

Gearboxes are common mechanical equipment that is used to change the rotation speed and transmit power. Since the working conditions of gearboxes are very poor, they are vulnerable to damage and failure. The fault diagnosis of a gearbox is one of the most important issues since people are concerned with maintaining equipment [1]. The vibration signals of a gearbox caused by gear impact, resonation, or mutual interference are very complex in actual working conditions. It is difficult to apply the conventional test and diagnosis methods effectively. The faulty characteristics for the diagnosis of a gearbox is different for different sizes of the cracking tooth, rotation speed, misalignment of gears, and so on [2–4]. Many researchers have established dynamical simulation models to analyze gear faults [5]. Brethee et al. [6] established an 18-degree-of-freedom vibration system to study the

different degrees of tooth-surface wear. However, their analytical equations are very complex and inconvenient to use.

A helical gearbox system is widely used in industrial machinery, and its faults directly cause large mechanical equipment to stop service and increase productivity costs. Early identification of faults in a helical gearbox is helpful for preventing equipment failure to avoid catastrophic losses. The types of gear fault include tooth surface pitting and wear [7–12], fatigue and crack [13–16], which can be described by measuring the vibration acceleration response of the gearbox. The spectrum of the response is characterized by the different levels of amplitude modulation and frequency modulation.

When a certain degree of shaft deflection or gear wear occurs, the impacted component may also be visible in the gear's vibration signal. The meshing frequency, modulation signal, and side-band information are very important features for the gear vibration signal [17]. Therefore, during the fault diagnosis, the impacted component and the frequency interval of the side-band can be used as the main basis for judging whether a fault happens in the gear system. When a gear failure that has the frequency bandwidth characteristics is measured by an acceleration sensor, high noise is distributed throughout the frequency spectrum. As a result, the frequency bands overlap with each other. Vibration signals also contain high-energy background signals related to the rotational speed.

When a gearbox fails, the actual measured signals are generally non-stationary and non-Gaussian signals, which usually contain strong noise. Therefore, it is difficult to eliminate the background signals and noise signals retaining fault information when using only conventional filtering methods. Wang et al. [18] presented an optimal demodulation subband selection method that can detect a gear fault by using envelope analysis, but the subband frequency features are not obvious in the environment with different noise levels.

Feature extraction of the vibration signal plays important roles in the health monitoring and fault diagnosis of rotating machinery. Empirical mode decomposition (EMD) was originally proposed by Huang [19] as an adaptive signal processing method that decomposes a complicated signal into a set of IMFs and a final residue. The IMFs only depend on the signal itself and have physical significance. The EMD can separate the nonlinear and non-stationary signal into several mode components. Parey [20] used a dynamic model of a spur gear for the early detection of a localized tooth with the EMD method. EMD has been successfully used in the fields of health monitoring, fault diagnosis, and detection [21]. However, there are some problems with the application of EMD. The EMD technique lacks the support of a theoretical foundation and has drawbacks of boundary effects, mode mixing, and stop effects in the signal processing.

The local mean decomposition (LMD) method proposed by Smith [22] uses a similar processing technique to EMD. LMD can decompose a multi-component signal into a series of mono-components that are defined by product functions (PFs). Each PF is obtained from the product of an envelope signal and a purely frequency-modulated signal. The LMD method avoids the overshooting and undershooting effect by using moving local means instead of cubic spline interpolation and improves upon the accuracy of the EMD method. However, the LMD method still has the problems of mode mixing [23] and time-consuming calculation [24].

The wavelet transform (WT) [25–27] has advantages of higher resolution and time-frequency information and is widely applied to the fault diagnosis of the rotating machinery. WT provides good results for analyzing noisy and non-stationary signals by selecting a suitable wavelet-based function. But the WT is non-adaptive decomposition method, which limit its applications, and the decomposition performance heavily degrades in a strongly noisy environment. Based on the theoretical framework of WT and the adaptivity of EMD, the empirical wavelet transform (EWT) was proposed by Gilles [28]. EWT can adaptively establish an optimum spectrum segmentation to decompose complex vibration modes according to the segmentations of the Fourier power spectrum. Kedadouche et al. [29] showed that the EWT technique can avoid the mode-mixing problem and has better performance than EEMD.

In practical applications, researchers need to give more attention to manufacturing many types of the faulty gears in order to increase the knowledge of fault information in the time domain,

frequency domain, or time-frequency domain by experimental measurements. In this case, Automatic Dynamic Analysis of Mechanical Systems (ADAMS) is used as a multibody dynamic simulation tool for a faulty gearbox system. The gearbox system model can be established using the Euler–Lagrange method in ADAMS software. The spectral subtraction method has been widely used in speech signal processing [30], but it can also be applied to gearbox fault diagnosis to remove partial additive noise. The EWT technique is adopted in this present work to determine the segment boundaries accurately and improve the fault-detection performance.

This paper is organized as follows. The spectral subtraction method and EWT method are introduced in Section 2. The experiment analysis is carried out in Section 3. The dynamical model of the gearbox system was established with ADAMS software, and the simulation verification of the proposed method is presented in Section 4. Finally, conclusions are presented in Section 5.

## 2. The Theory of the Proposed Method

A new signal-processing scheme is proposed to detect gearbox faults at constant speed. A flow chart of the fault diagnosis method is shown in Figure 1. To improve the fault information, the spectral subtraction technique is firstly used to remove the partial noise of the signal. The impacted component of the gear fault signal is obtained with the EWT method, and then the side-band frequency of the fault features is obtained with the square envelope spectrum method.

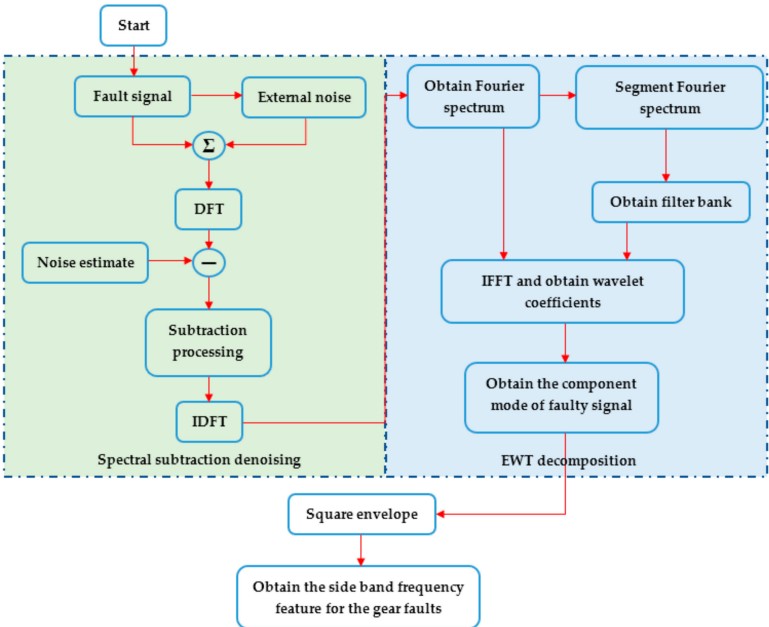

**Figure 1.** Flow chart of the fault diagnosis method.

### 2.1. Spectral Subtraction Method

The spectral subtraction method is widely employed in signal processing for speech enhancement in order to remove acoustic noise [31–33]. Spectral subtraction is usually used as a denoising method for fault detection. The proposed technique is appropriate for the fault diagnosis of rotating machinery in steady state at constant speed. It is only applied to stationary signals, which means the frequency content is independent of time. In this context, a new denoising scheme is employed to enhance the useful information of the vibration signals measured from a faulty gearbox. It is assumed that the vibration signal is disturbed by random noise, which can be removed by subtracting of an estimation of the mean of the noise spectrum from the noisy signal spectrum. The measured signal model in the time domain is given by:

$$z(m) = s(m) + n(m) \tag{1}$$

where $z(m)$ is the measured mixed-signal (including the vibration signal and noise signal), $s(m)$ is the vibration signal of the gear, $n(m)$ is the additive noise, and $m$ is the discrete-time index. The frequency-domain representation of the relation is given by:

$$Z(k) = \mathcal{F}\{z(m)\} = S(k) + N(k) \tag{2}$$

where $Z(k)$, $S(k)$, and $N(k)$ are the discrete Fourier transforms of $z(m)$, $s(m)$, and $n(m)$, respectively. $k$ is the discrete-frequency index.

In the spectral subtraction method, the estimation of the original signal power spectrum can be obtained with the variation of parameters. It includes an over-subtracting estimate of the noise power spectrum and a lower limiting value for the resulting spectrum. The form of the enhanced signal power spectrum is:

$$\left|\hat{S}(k)\right|_2 = \max\left\{\left|Z(k)\right|_2 - \alpha\left|\hat{N}(k)\right|_2, \beta\left|\hat{N}(k)\right|_2\right\} \tag{3}$$

where $\alpha$ is the over-subtracting multiplication factor, which has a value greater than or equal to 1 and determines the degree of distortion of the fault signal. $\beta$ is the spectral flooring parameter, which has a value between 0 and 1, and controls the effect of residual noise. The multiplication factor $\alpha$ in Equation (3) is defined by the function of the segmental signal-noise ratio (SNR), which is estimated in the frame and defined by:

$$\alpha = \begin{cases} 5 & SNR < -5 \\ 4 - \frac{3}{20}SNR & -5 \leq SNR \leq 20 \\ 1 & SNR > 20 \end{cases} \tag{4}$$

$$SNR = 10 \log 10 \left(\frac{\sum\limits_{k=0}^{L-1}\left|Z(k)\right|^2}{\sum\limits_{k=0}^{L-1}\left|N(k)\right|^2}\right) \tag{5}$$

Finally, the enhanced signal is obtained by the inverse discrete Fourier transform (IDFT):

$$\hat{s}(m) = \mathcal{F}^{-1}\left\{\left|\hat{S}(k)\right|e^{j\phi(k)}\right\} \tag{6}$$

where $\phi(k)$ is the phase function of the DFT of the input signal.

According to the basic theory of spectral subtraction, the vibration noise is estimated over the whole spectrum by adjusting the multiplication factor $\alpha$ [34]. By using the spectral subtract technique, the faulty signal is enhanced. However, it cannot extract the mono-component information. To identify the side-band frequency feature, EWT is used to analyze the faulty signal.

### 2.2. Empirical Wavelet Decomposition

EWT is a new signal-processing algorithm to detect the different vibration modes based on the EMD method and wavelet analysis theory. It can effectively extract the different modes from a mixed vibration signal, by adaptively establishing an appropriate filter bank based on the Fourier spectrum. EMD can adaptively decompose a signal into IMFs and obtain a time-frequency resolution to extract the fault features, but it is lacking the supporting mathematical theory. In contrast, EWT has no frequency resolution, but also a clear mathematical theoretical basis and the high computation speed of the discrete WT. The EWT procedure is described as follows:

Firstly, accordingly to the Shannon criteria, the frequency limit is defined as $[0, \pi]$ in a normalized Fourier axis of periodicity $2\pi$. The initial Fourier support $[0, \pi]$ is segmented into N contiguous segments containing the spectra of definite modes. The boundary of each segment is denoted as $\omega_n$, and hence, the bandpass filter is obtained: $\Lambda_n = [\omega_{n-1}, \omega_n]$. In order to realize the EWT on the

segmental support $\Lambda_n$, the transition phase $T_n$ of width $2\tau_n$ is defined as being centered on each $\omega_n$. An empirical filter bank is provided in Figure 2. The simplest $\tau_n$ is proportional to $\omega_n$: $\tau_n = \gamma\omega_n$, where the range of $\gamma$ is $0 < \gamma < \min_n \frac{\omega_{n+1}-\omega_n}{\omega_{n+1}+\omega_n}$.

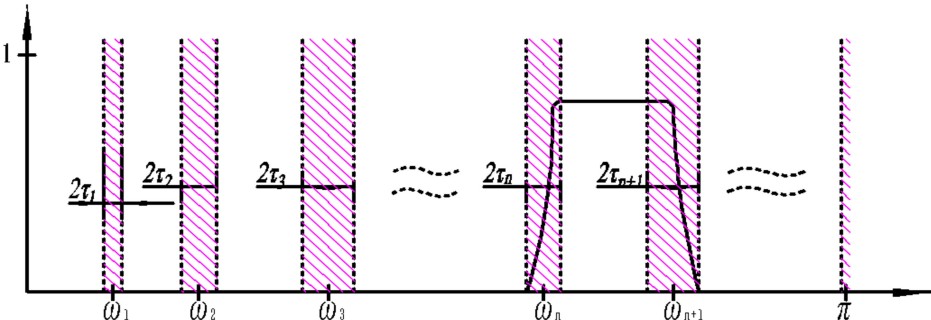

**Figure 2.** Segmenting Fourier spectrum into N contiguous segments.

Next, the empirical wavelets are defined as bandpass filters on each Fourier support $\Lambda_n$. The empirical scaling function and empirical wavelets are established using the idea of the construction of both Littlewood–Paley and Meyer's wavelets, respectively. The simplified form is shown:

$$\hat{\phi}_n(\omega) = \begin{cases} 1 & |\omega| \leq (1-\gamma)\omega_n \\ \cos[\frac{\pi}{2}\beta(\frac{1}{2\gamma\omega_n}(|\omega|-(1-\gamma)\omega_n))] & (1-\gamma)\omega_n \leq |\omega| \leq (1+\gamma)\omega_n \\ 0 & otherwise \end{cases} \tag{7}$$

$$\hat{\psi}_n(\omega) = \begin{cases} 1 & (1+\gamma)\omega_n \leq |\omega| \leq (1-\gamma)\omega_{n+1} \\ \cos[\frac{\pi}{2}\beta(\frac{1}{2\gamma\omega_{n+1}}(|\omega|-(1-\gamma)\omega_{n+1}))] & (1-\gamma)\omega_{n+1} \leq |\omega| \leq (1+\gamma)\omega_{n+1} \\ \sin[\frac{\pi}{2}\beta(\frac{1}{2\gamma\omega_n}(|\omega|-(1-\gamma)\omega_n))] & (1-\gamma)\omega_n \leq |\omega| \leq (1+\gamma)\omega_n \\ 0 & otherwise \end{cases} \tag{8}$$

where $\beta(x)$ is an arbitrary function with an independent variable $x$:

$$\beta(x) = \begin{cases} 0 & x \leq 0 \\ x^4(35 - 84x + 70x^2 - 20x^3) & 0 < x < 1 \\ 1 & x \geq 1 \end{cases} \tag{9}$$

According to the theoretical framework of the wavelet transform, the empirical wavelet transform is built by the constructed bandpass wavelet filter bank. The detail coefficient of the EWT is the inner product between the analyzed signal and the empirical wavelet in the time domain, and is the inverse Fourier transform of the of the frequency function production:

$$W_s^\varepsilon(n,t) = \langle s(t), \psi_n(t) \rangle = \int s(\tau)\overline{\psi_n(\tau-t)}d\tau = \mathcal{F}^{-1}\left(\hat{S}(\omega)\overline{\hat{\psi}_n(\omega)}\right) \tag{10}$$

The approximation coefficient of the EWT is the inner product between the analyzed signal and the scaling function in the time domain, and is the inverse Fourier transform of the frequency function production:

$$W_s^\varepsilon(0,t) = \langle s(t), \phi_1(t) \rangle = \int s(\tau)\overline{\phi_n(\tau-t)}d\tau = \mathcal{F}^{-1}\left(\hat{S}(\omega)\overline{\hat{\phi}_n(\omega)}\right) \tag{11}$$

Finally, the mono-component modes are obtained through the details and approximation coefficient of the EWT. The construction of the original signal is obtained by:

$$s(t) = W_f^\varepsilon(0,t) * \phi_1(t) + \sum_{n=1}^{N} W_f^\varepsilon(n,t) * \psi_n(t) = \mathcal{F}^{-1}\left( \hat{W}_f^\varepsilon(0,\omega)\hat{\phi}_1(\omega) + \sum_{n=1}^{N} \hat{W}_f^\varepsilon(n,\omega)\hat{\psi}_n(\omega) \right) \quad (12)$$

The mono-component modes are given by:

$$s_0(t) = W_s^\varepsilon(0,t) * \phi_1(t) \quad (13)$$

$$s_k(t) = W_s^\varepsilon(k,t) * \psi_k(t) \quad (14)$$

*2.3. The Proposed SS-AEWT for Extracting Gear Fault Features*

If the fault information can be enhanced by using the spectral subtraction method, the mono-component mode of the faulty signal is clearly obtained by EWT. The mono-component mode is described by:

$$L(m) = \hat{s}(m) * \Gamma \quad (15)$$

where $\hat{s}(m)$ denotes the estimated value of the faulty signal obtained by the spectral subtraction method, and $\Gamma$ represents the decomposition band width of EWT.

Envelope analysis is an effective technique for the fault diagnosis of rotating machinery, and it has the advantage of simple application and low computational effort. This algorithm can demodulate a multi-component signal to obtain a representation of the mono-component signal for a cyclostationary impulse signal. In this paper, the side-band frequency features are extracted by squared envelope spectrum (SES) analysis.

## 3. Experiment Analysis

In this subsection, the experimental data of CETIM (Centre des Etudes Techniques des Industries Mécaniques de Senlis) [35] is used to extract the side-band frequency features with the proposed method. The experimental device includes a motor, brake, gearbox, and bearing. The single-stage reducer for the gearbox includes a pinion with 20 teeth and a gear with 21 teeth. The input rotation speed is 1000 rpm, and the sampling frequency is 20 kHz. The spectral subtraction denoising method is applied to obtain the processed signal and compare it to the original signal. Figure 3 shows that the impulse response of the fault information is much clearer.

The frequency spectrum results are compared in Figure 4. The gear meshing frequency is clearer than the side-band frequency in the frequency spectrum of the original signal, as shown in Figure 4a. However, the side-band frequency features are enhanced by the spectral subtraction method, as shown in Figure 4b. The results indicate that the noise can be effectively removed by the spectral subtraction method to obtain the processed signal, which is beneficial for extracting the fault features by the EWT method.

The proposed method can extract a greater amplitude of the side-band frequency for the gear fault feature. Figure 5 and Table 1 show the detailed information. The amplitude values of the first and second-order frequency are better for the proposed method than the others. The amplitude values of the first-order frequency are improved by 36.26%, 45.52%, and 43.89% with EMD, LMD, and DWT, respectively. The amplitude value is not improved by more than 50%, because the external noise is very low. Furthermore, the calculated frequency is 16.17 Hz, which approximately equals the theoretical value (15.87 Hz). Hence, we can conclude that the first-order frequency in the side-band frequency features equals the rotation frequency of the input shaft. This demonstrates that the proposed method is effective for gearbox fault diagnosis.

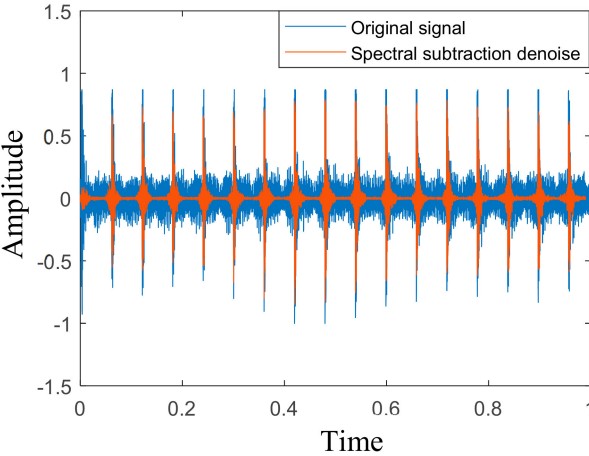

**Figure 3.** The compared results with spectral subtraction denoising.

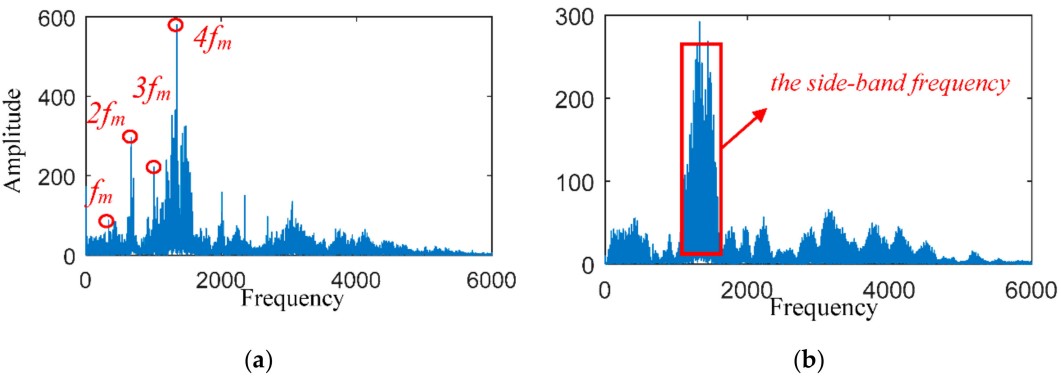

(**a**)　　　　　　　　　　　　　　　　　　　　　　　　(**b**)

**Figure 4.** Frequency spectrum results with spectral subtraction denoising: (**a**) frequency spectrum of the original signal; and (**b**) frequency spectrum of the processed signal.

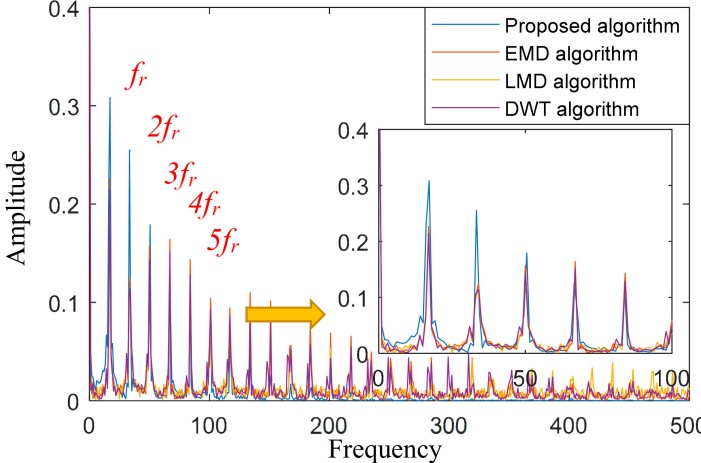

**Figure 5.** The fault features with square envelope analysis.

**Table 1.** The side-band amplitudes for different analysis methods.

| Analysis Type | Amplitude of fr | Amplitude of 2fr | Amplitude of 3fr | Amplitude of 4fr | Amplitude of 5fr |
|---|---|---|---|---|---|
| EMD algorithm | 0.2264 | 0.1252 | 0.1572 | 0.1639 | 0.1435 |
| LMD algorithm | 0.212 | 0.1154 | 0.1341 | 0.1468 | 0.1245 |
| DWT algorithm | 0.2144 | 0.1147 | 0.1429 | 0.1515 | 0.1287 |
| Proposed algorithm | 0.3085 | 0.255 | 0.1789 | 0.1248 | 0.1203 |

## 4. Verification and Discussion of the Proposed Method

When gears break down, the faulty component often causes an impact, which generates impact energy. Types of gear fault include missing tooth, broken tooth, chipping tip, root crack, and spalling conditions [36–38]. However, in the case of a broken tooth, the gearbox needs to change to a new gear. Therefore, in order to investigate the fault features of the helical gear, a different degree fault of breakage gear is implemented in this paper [39]. The tooth width and broken width are defined as L and B, so the degree of breakage is B/L. The breakage size is shown in Figure 6.

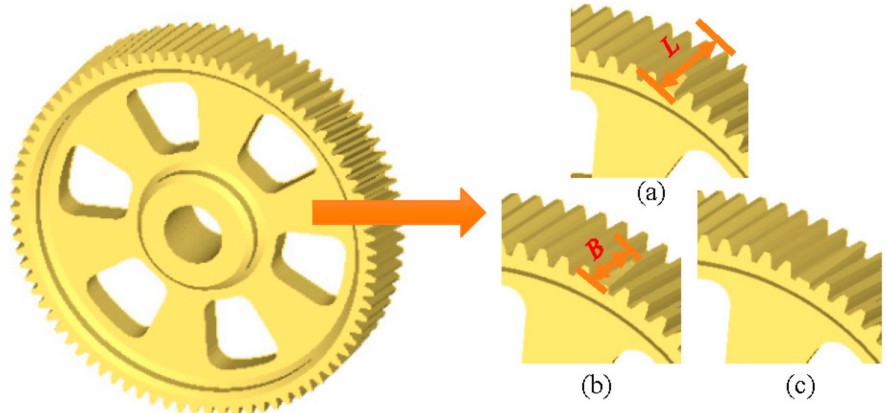

**Figure 6.** The degree of tooth breakage, (**a**) 1/3 breakage, (**b**) 2/3 breakage, and (**c**) total breakage.

### 4.1. Establishment of Simulation Model

ADAMS provides the measured data for the strategy of the faulty gear in various test environments, which are important for the modes and intensities of the monitored gears. Several steps are necessary before obtaining the simulation signal. Firstly, ADAMS applies gravity to all the parts and leads all the solid models (including the gearbox body, the gear shafts and gears) with material properties. Then the system automatically generates information about the mass of the components, the center of mass, and the rotational inertia of the components. To increase the accuracy of the dynamic model, the faulty gear is modeled as a flexible body using a finite element analysis tool. Finally, the independent components are joined by different types of constraints. In this paper, the reduction gearbox includes input shaft, middle shaft, output shaft, and two pairs of gears, and the number of teeth is 17, 81, 23, and 109, respectively. The dynamical model shows in Figure 7. The input rotation speed of the gearbox is set to 2880 RPM (48 Hz), and the sampling frequency is 20 kHz.

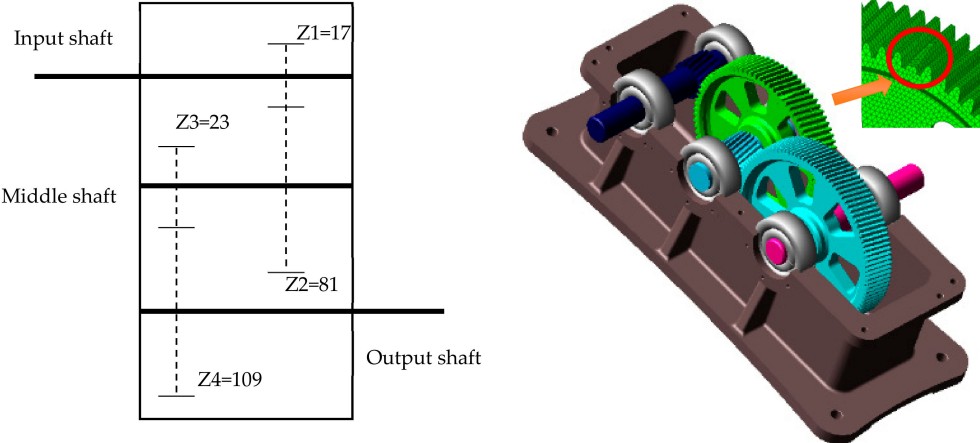

**Figure 7.** The configuration and dynamical model of the gearbox system.

### 4.2. The Fault Analysis of Different Breakages

In this subsection, different gear faults are analyzed using the dynamical model of the gearbox system, and side-band frequency features are obtained by using different analytical methods. The analytical results are shown in Figure 7.

According to the parameters of the gearbox, the rotational frequency of the faulty gear shaft is $f_r = f_{in}(Z_1/Z_2) = 48Hz * (17/81) = 10.07Hz$. The first-order frequency value of the analytical result is 10.1 Hz, which is close to the theoretical value. The variation of each order frequency amplitude is consistent with the experimental data, the comparative results are shown in Figures 5 and 8b,c. Hence, the dynamical model can reflect the fault characteristics of the tooth breakages. The proposed method can still clearly extract the side-band frequency feature with the 1/3 tooth breakage, but other methods have lost the ability to detect the gear fault. The result is shown in Figure 8a.

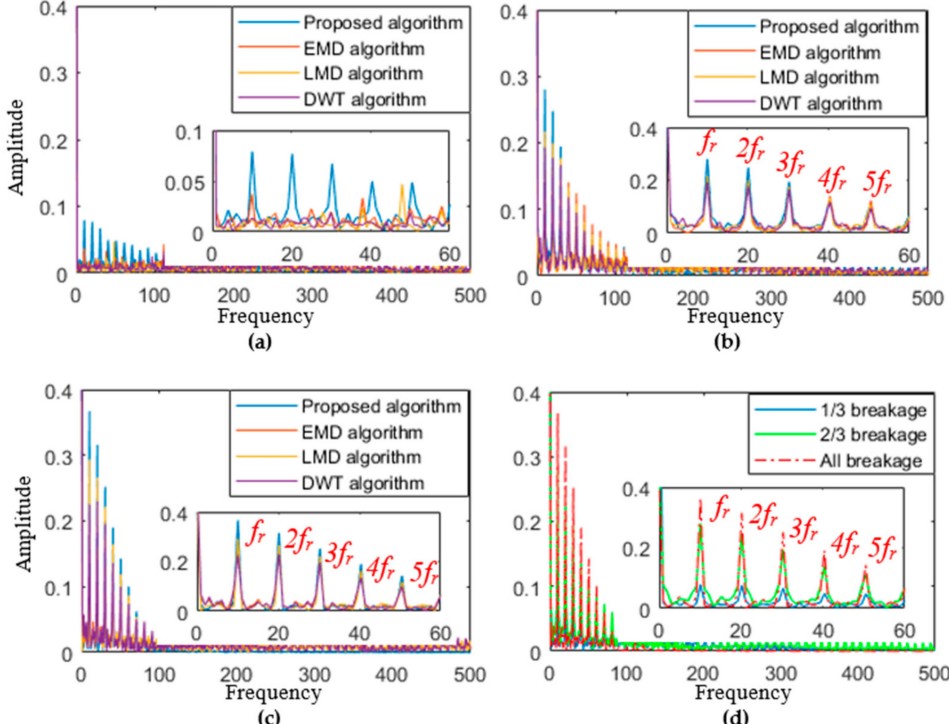

**Figure 8.** The results comparison (**a**) 1/3 breakage, (**b**) 2/3 breakage, (**c**) total breakage, (**d**) the proposed method with different breakage.

The results from Table 2, the amplitude value of frequency feature for EMD method is much bigger than LMD and DWT method with the different tooth breakage. However, the amplitude value obtained by the proposed method is 53%, 45%, and 32% higher than that of EMD with the different tooth breakage, respectively. Especially 1/3 tooth breakage, other methods are very difficult to detect the side-band frequency features. The results indicate that the proposed method can clearly extract the side-band frequency features without the external noise, and the amplitude value is enlarged with increasing tooth breakage.

**Table 2.** Comparison of the first-order frequency amplitude value of different analysis methods.

| Fault Type | EMD | LMD | DWT | Proposed Algorithm | Percentage |
|---|---|---|---|---|---|
| 1/3 breakage | 0.037 | 0.011 | 0.013 | 0.079 | 53%, 86%, 84% |
| 2/3 breakage | 0.22 | 0.24 | 0.21 | 0.32 | 45%, 33%, 52% |
| Total breakage | 0.25 | 0.29 | 0.22 | 0.37 | 32%, 22%, 41% |

### 4.3. The Fault Analysis of the SNR Effect

The detection of the side-band frequency was applied to the simulated signal for 2/3 tooth breakage with different levels of noise added. Because there is no external noise, the simulated signal for the faulty gear is very a clear impulse signal, as shown in Figure 9a. However, the vibration signal is weak for an early fault diagnosis of a gearbox and often buried under background noise. Next, the simulated signal with different white noise (10 dB, 5 dB, 0 dB, −5 dB, and −10 dB) was processed by the proposed method. The waveforms are shown in Figure 9b–f in the time domain. The impulse signals become increasingly misty as the SNR decreases, especially with noise under 0 dB.

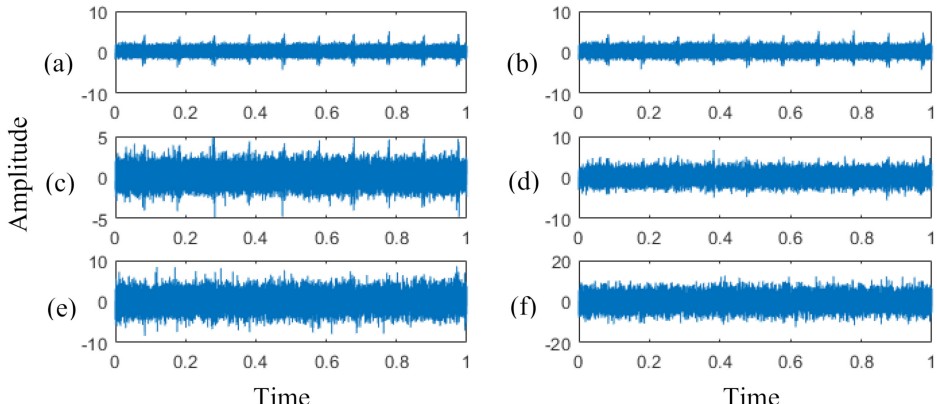

**Figure 9.** The simulated signal with added noise, (**a**) no noise, (**b**) 10 dB, (**c**) 5 dB, (**d**) 0 dB, (**e**) −5 dB, and (**f**) −10 dB of noise.

The corresponding waveforms and envelope spectra of the faulty component information are extracted by using EWT and a square envelope, respectively. The results are shown in Figures 10 and 11. The fault feature information is basically reserved above −5 dB of noise, and the side-band components are obtained by envelope analysis. This indicates that the frequency feature of the faulty component of a gear is effectively extracted by the proposed method. The fundamental frequency equals the result of the theoretical calculation, and the amplitude value of the order frequency remains stable.

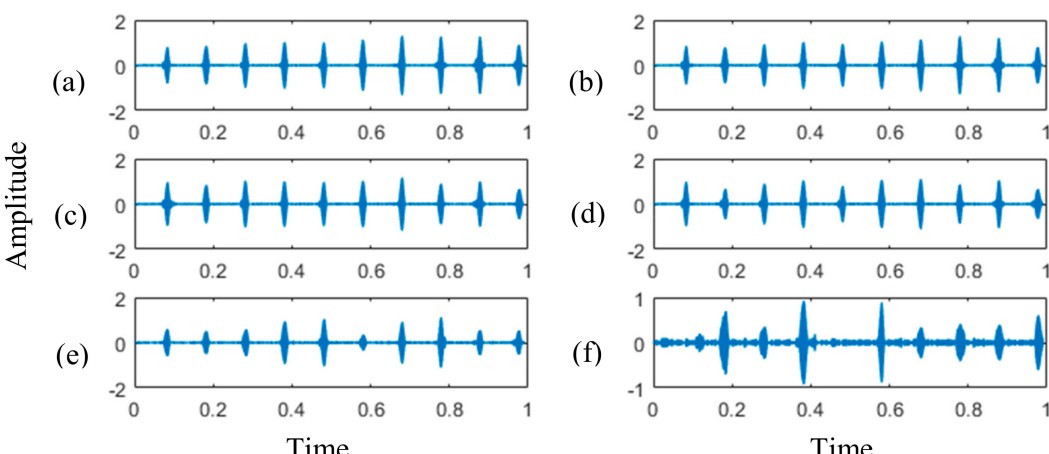

**Figure 10.** The impulse component with EWT method, (**a**) no noise, (**b**) 10 dB, (**c**) 5 dB, (**d**) 0 dB, (**e**) −5 dB, and (**f**) −10 dB of noise.

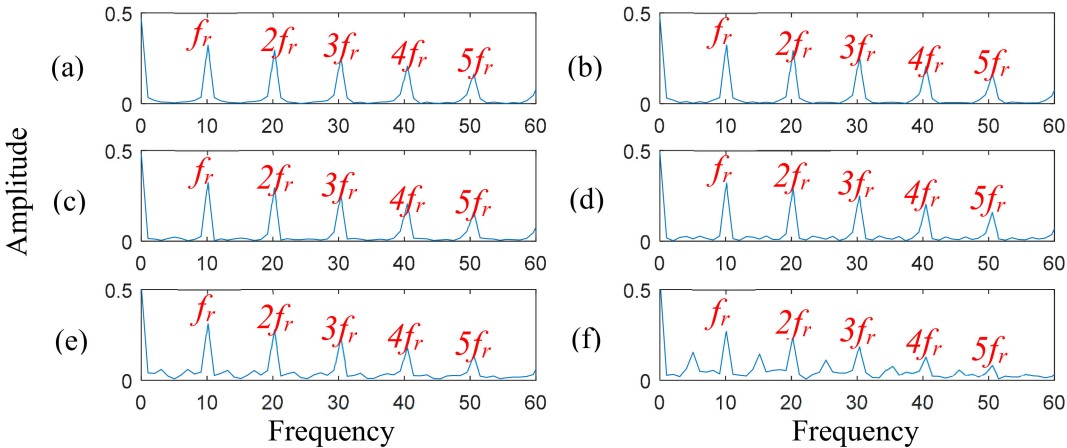

**Figure 11.** The faulty features with square envelope analysis, (**a**) no noise, (**b**) 10 dB, (**c**) 5 dB, (**d**) 0 dB, (**e**) −5 dB, and (**f**) −10 dB of noise.

In Figure 12, the proposed signal processing can effectively extract the side-band frequency component, and stably obtain the fixed amplitude value where the SNR level is low. As a result, the proposed method indicates that it can largely enhance the representation of the faulty gear component and make the amplitude of the side-band frequency more stable under the strong noise.

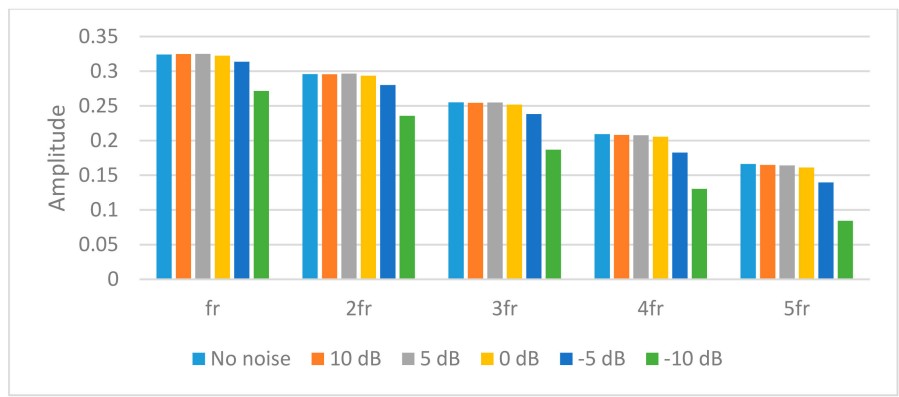

**Figure 12.** The comparison results for different noise level.

*4.4. Comparison of Different Methods*

To verify the effectiveness of the proposed method, the simulated signal with 5 dB of noise was considered on the 2/3 tooth breakage situation. The frequency spectrum is obtained with the spectral subtraction method. Figure 13 shows that the side-band frequency under the strong noise is hardly extracted from the frequency spectrum. However, the spectral subtraction method can enhance the amplitude value of the side-band frequency region more than the gear meshing frequency region. The results are shown in Figure 14. The side-band frequency features are extracted by the EWT method.

For comparison, the EMD was used to decompose the observed signal into eight IMFs. The added noise signals are complex, so the fault frequency cannot be evaluated directly with the power spectrum. The envelope spectrum of the first IMF contains the primary information of the fault signal. Therefore, the side-band feature frequency can be identified clearly. Then, the LMD is applied to adaptively decompose the added noise signal into several PFs from the high-frequency band to the low one. Several PFs can be obtained with the LMD process. The first PF is selected for further analysis, because it has the biggest correlation coefficient and keeps the most of information from the original signal. Finally, the DWT decomposes the added noise signal to extract the main information of the fault features. The side-band frequency features are extracted from the wavelet correlation of the impulse

component with the envelope spectrum. The compared faulty features of the different algorithms are shown in Figure 15.

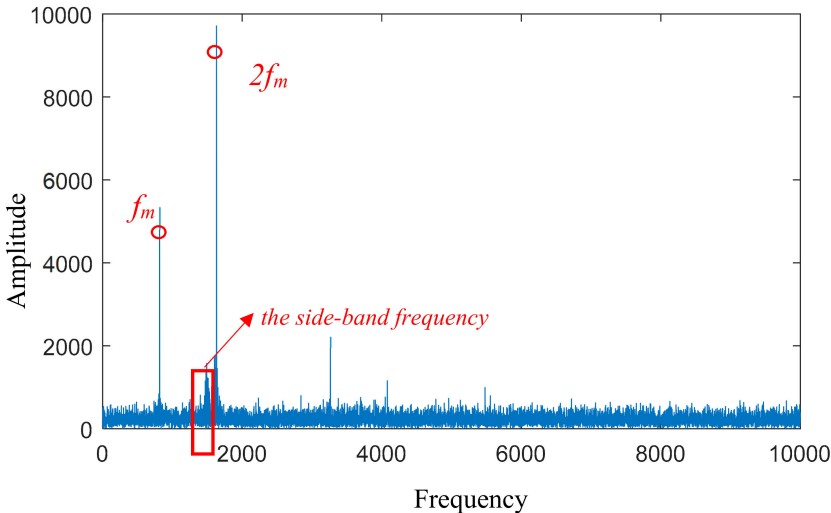

**Figure 13.** The segmentation of the frequency spectrum with 5dB of noise.

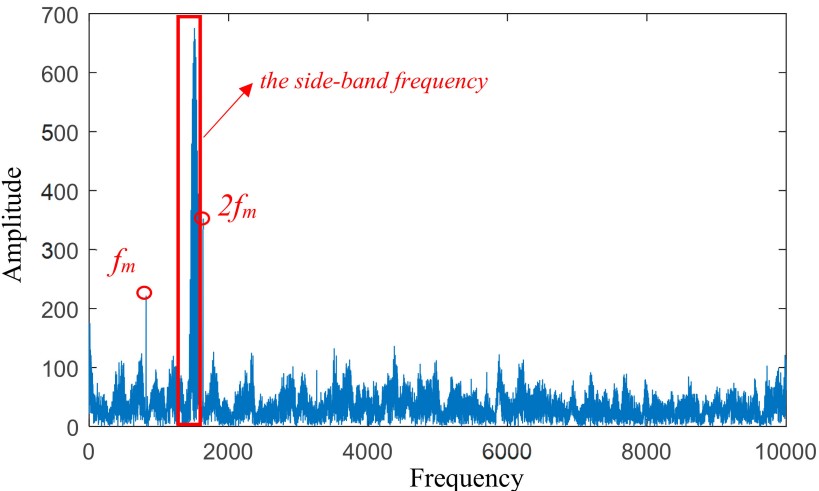

**Figure 14.** The segmentation of the frequency spectrum with spectral subtraction.

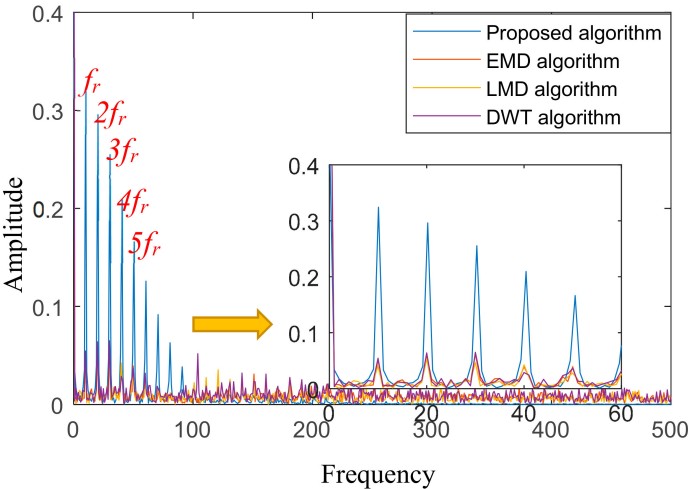

**Figure 15.** The fault features with square envelope analysis.

Table 3 shows the side-band features calculated from the square envelope analysis. The proposed algorithm has significantly better performance for the feature extraction of the side-band information of the faulty gear. The noise energy has been obviously reduced by the proposed method, which is efficient for fault feature extraction.

**Table 3.** The side-band amplitudes for different analysis method.

| Analysis Type | Amplitude of fr | Amplitude of 2fr | Amplitude of 3fr | Amplitude of 4fr | Amplitude of 5fr |
|---|---|---|---|---|---|
| EMD algorithm | 0.051 | 0.0595 | 0.0619 | 0.0392 | 0.0393 |
| LMD algorithm | 0.0402 | 0.0533 | 0.0465 | 0.0425 | 0.0279 |
| DWT algorithm | 0.0546 | 0.0639 | 0.0651 | 0.0288 | 0.0365 |
| Proposed algorithm | 0.3251 | 0.2965 | 0.2548 | 0.2078 | 0.1642 |

Table 4 shows that EMD, LMD, and DWT method can extract the side-band frequency features except noise −5dB and −10dB. Their performance for extracting the side-band feature reached a limit when the noise exceeded −5dB. However, the proposed method can more clearly extract the side-band feature under different background noise, and its amplitude value remains stable. Therefore, the proposed method is feasible for reducing strong noise and effectively extracting the faulty features.

**Table 4.** The amplitude value of first-order frequency for different analysis methods and SNR.

| SNR | EMD | LMD | DWT | Proposed Algorithm | Percentage |
|---|---|---|---|---|---|
| 10 dB | 0.0965 | 0.0518 | 0.0675 | 0.3247 | 236%, 527%, 381% |
| 5 dB | 0.0510 | 0.0401 | 0.0351 | 0.3251 | 537%, 711%, 826% |
| 0 dB | 0.0266 | 0.0343 | 0.0156 | 0.3225 | 1112%, 840%, 1967% |
| −5 dB | 0.0109 | 0.0138 | 0.0067 | 0.3136 | 2777%, 2172%, 4581% |
| −10 dB | 0.0036 | 0.0016 | 0.0063 | 0.2716 | 7444%, %, 4211% |

## 5. Conclusions

This paper has proposed the adaptive EWT technique combined with the spectral subtraction method for fault diagnosis of a gearbox system. Firstly, the side-band frequency features are extracted to verify the effectiveness of the algorithms with the experimental data when the gear breaks down. The results indicate that the performance of the proposed method is better than other methods. Then the multi-body dynamics model of the gearbox system is established after gear flexibility treatment. The different tooth breakages are designed by the dynamical model. The experimental results indicate that the dynamical model can satisfy the gear fault characteristics, and it can be used to validate the algorithm performance. For smaller tooth breakage, the proposed method can more clearly detect the fault features than EMD, LMD, and DWT techniques. When the fault features are not extracted by EMD, LMD, and DWT techniques at a noise above −5dB, the proposed method can still discern the side-band frequency features.

The spectral subtraction technique can eliminate the strong noise to enhance the faulty component. Then, the EWT method can effectively extract the transient impulse component, and the side-band frequency features are significantly obvious. The multi-body dynamics model of the gearbox system after gear flexibility treatment is helpful for establishing the gearbox fault. Furthermore, it can be used to design multiple faults of the gearbox and provide much data for predicting gearbox faults.

**Author Contributions:** C.-M.L. gave academic guidance to this research work and revised the manuscript. P.W. designed the core methodology of this study, programmed the algorithms and carried out the experiments, and drafted the manuscript. Both authors read and approved the final manuscript.

**Funding:** This work was supported by a 2017 grant from the Russian Science Foundation (Project No. 17-19-01389).

**Conflicts of Interest:** The authors declare no conflict of interest.

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
