# Peer review of "Fault Diagnosis of a Helical Gearbox Based on an Adaptive Empirical Wavelet Transform in Combination with a Spectral Subtraction Method"

_applsci, doi:10.3390/app9081696_

Round 1
Reviewer 1 Report
Authors proposes a spectral subtraction denoising method combined with adaptive empirical wavelet transform to diagnose the fault characteristics for a helical gearbox. My suggestions are here:
Please revise the abstract in where a link between the first and second sentences is missing. Author should check and revise the format of abstract as well as entire manuscript.
Authors mentioned ‘simulation signal’ in Fig. 1 which is not readable. The presentation of block diagram is too poor.
Authors should revise Left hand side part of fig.2 which is not readable.
Why Authors set the rotation speed 2880rpm (48Hz) instead of 3000rpm (50Hz)? Why the sampling frequency (20KHz) is set to 20KHz which is too high?
Authors just presented some sorts of simulated results without explaining it's effects in details which is not sufficient to put the idea going forward.
Why the rotation speed of experimental results is set 1000rpm which does not match the simulated one?
The experimental results are insufficient. How Authors claimed that experimental results verified the simulated results?
Author Response
Dear Reviewers:
Thank you for your letter and for the reviewers’ comments concerning our manuscript entitled “Fault diagnosis of a helical gearbox based on an adaptive empirical wavelet transform in combination with a spectral subtraction method” (ID: 456753). Those comments are all valuable and very helpful for revising and improving our paper, as well as the important guiding significance to our researches. We have studied comments carefully and have made the correction which we hope to meet with approval. Revised and deleted portion are marked in red and blue in the paper, respectively. The main corrections in the paper and the responds to the reviewer’s comments are as following:
Responds to the reviewer’s comments:
Point 1: Please revise the abstract in where a link between the first and second sentences is missing. Author should check and revise the format of abstract as well as entire manuscript.
Response 1: We have re-written this part according to the Reviewer’s suggestion. A link word has been added between the first and second sentence. In addition, the abstract and manuscript have been checked and revised as well.
Point 2: Authors mentioned ‘simulation signal’ in Fig. 1 which is not readable. The presentation of block diagram is too poor.
Response 2: Thank you for your suggestion. Figure 1 is revised.
Point 3: Authors should revise Left hand side part of fig.2 which is not readable.
Response 3: Thank you for your careful reviewing. Figure 2 is revised.
Point 4: Why Authors set the rotation speed 2880rpm (48Hz) instead of 3000rpm (50Hz)? Why the sampling frequency (20KHz) is set to 20KHz which is too high?
Response 4: Frankly speaking, there is no influence on the performance of the proposed algorithm no matter what the rotation speed is, as long as the RPM is in the reasonable range when a gearbox working. In this manuscript, the real experimental signal comes from CETIM (Centre des Etudes Techniques des Industries Mécaniques de Senlis), in which the sampling frequency is originally set as 20KHz. When doing the frequency spectrum analysis, the data length is set as 40001 points, so that the frequency resolution is accurate enough to distinguish the difference of the failure characteristics. Therefore, the 20KHz sampling frequency has no influence on the test results and its effectiveness.
Point 5: Authors just presented some sorts of simulated results without explaining it's effects in details which is not sufficient to put the idea going forward.
Response 5: Thank you for your suggestions. Your comments are very important for us to improve the quality of this manuscript. Some newly designed simulation tests have been carried out, and the test results are presented by the tables and figures. Then, the performance of the proposed algorithm working in different conditions are validated, and the advantages of the proposed algorithm over other existed algorithms are discussed in details. The further conclusion is drawn, in which the proposed algorithm has better performance than others when working in the strong noise conditions and the failure characteristic is not obvious. The newly adding tables and figures can better support our idea about the proposed algorithm.
Point 6: Why the rotation speed of experimental results is set 1000rpm which does not match the simulated one?
Response 6: The experimental signal is from the CETIM, which was often used to test the effectiveness of the new algorithms. In this manuscript, the effectiveness of the proposed algorithm is validated by using existed CETIM experimental signal firstly. Then, the performance of the proposed algorithm compared to other existed algorithms are investigated based on the simulation experiments. To be honest, we are not intended to prove how much the simulation model would match the real experiment in the parameters. In fact, the simulated model is an efficient approach to test the performance of a new algorithm working in the different conditions, in which the variables would be limited and the model is easy to be modified. However, although the RPM in the experimental signal is different from the RPM in the simulation signal, the signal changing regularity is very consistent. Therefore, this simulation model would be used in the development of new algorithms.
Point 7: The experimental results are insufficient. How Authors claimed that experimental results verified the simulated results?
Response 7: The manuscript has been re-organized, and more simulation experiments are carried out to support our idea about the proposed algorithm. Firstly, the existed experimental signal of CETIM is used to test the effectiveness of the proposed algorithm. Then, the simulation signal is used to investigate the performance of the proposed algorithm compared to the other existed algorithm. Although the real experimental result is not sufficient to prove the advantages of the proposed algorithm over the others, the simulation experiments would be helpful to test the performance in the different conditions. Finally, the test results show that the proposed algorithm has obvious advantages over other existed algorithms no matter in the strong noise conditions or small damage conditions.
Special thanks to you for your good comments.
We tried our best to improve the manuscript and made some changes in the manuscript. Here we did not list the changes but marked in red in revised paper.
We appreciate for Editors/Reviewers’ warm work earnestly and hope that the correction will meet with approval.
Once again, thank you very much for your comments and suggestions.

Reviewer 2 Report
The paper deals with
Fault diagnosis of a helical gearbox based on an adaptive empirical wavelet transform in combination 2 with a spectral subtraction method.
In spite of the incontestable appeal of the proposed approach there are few points, which impair its quality:
1) A formal definition of a fault has to be provided along with its interpretation for the gearbox.
2) The quality of equations is very low, and hence, their readability is decreased.
3) Formulas (1)-(18) are very classical and I do not really see the contribution of the paper.
4) As far as I understood, the presented results concern simulations using ADAMS software. I am not very convinced if this allows deliberations about real nature of faults present in the gearbox.
5) The simulation results should be presented in a more condensed form in favour of real experiments presented in Section 4.
Author Response
Dear Reviewers:
Thank you for your letter and for the reviewers’ comments concerning our manuscript entitled “Fault diagnosis of a helical gearbox based on an adaptive empirical wavelet transform in combination with a spectral subtraction method” (ID: 456753). Those comments are all valuable and very helpful for revising and improving our paper, as well as the important guiding significance to our researches. We have studied comments carefully and have made the correction which we hope to meet with approval. Revised and deleted portion are marked in red and blue in the paper, respectively. The main corrections in the paper and the responds to the reviewer’s comments are as following:
Responds to the reviewer’s comments:
Point 1: A formal definition of a fault has to be provided along with its interpretation for the gearbox.
Response 1: We cite some papers to illustrate the fault type of gear.
Point 2: The quality of equations is very low, and hence, their readability is decreased.
Response 2: We delete some unnecessary formulas and add a figure to explain the segments of the Fourier spectrum for EWT technique.
Point 3: Formulas (1)-(18) are very classical and I do not really see the contribution of the paper.
Response 3: Our aim is to reduce the noise and clearly identify the fault characteristics, the spectral subtraction method combining EWT technique is firstly presented in this paper. The results indicate that the proposed method is helpful for the fault diagnosis of the gearbox. In order to illustrate the algorithm process, we add a formula (15) and a detailed flow chart (Fig.1).
Point 4: As far as I understood, the presented results concern simulations using ADAMS software. I am not very convinced if this allows deliberations about real nature of faults present in the gearbox.
Response 4: Firstly, the algorithms are proved by the experimental data. The results indicate that the performance of the proposed method is better than other methods. Secondly, the simulated signal is obtained from the dynamical model with the tooth breakage, and its results are consistent with experimental results. Finally, discussing the limits of the proposed algorithm in different conditions. Hence, the dynamical model is effective and useful to verify the advantage of the algorithms.
Point 5: The simulation results should be presented in a more condensed form in favour of real experiments presented in Section 4.
Response 5: We add more experiments. In order to illustrate the advantage of the proposed method, this paper discusses the limits of the algorithm in different conditions (tooth breakage and strong noise). For the different tooth breakages, the simulated signal is used to verify the ability of the algorithms to extract the side-band features. The results indicate that the proposed method can clearly extract the side-band frequency with 1/3 tooth breakage, but other methods have lost effectiveness. For the strong background noise, the proposed method can more clearly extract the side-band frequency than other methods, and its amplitude value is more stable. Therefore, the dynamical model is helpful for the fault diagnosis of the gearbox in the future.
Special thanks to you for your good comments.
We tried our best to improve the manuscript and made some changes in the manuscript. Here we did not list the changes but marked in red in revised paper.
We appreciate for Editors/Reviewers’ warm work earnestly and hope that the correction will meet with approval.
Once again, thank you very much for your comments and suggestions.

Round 2
Reviewer 1 Report
Though the revised manuscript is well organised, the quality of accompanying letter is not maintain the standard. Authors should describe the detail of changes in accompanying letter mentioning section and page number. The experimental results are still insufficient which lacks the quality of the manuscript.
Author Response
Dear Reviewers:
Thank you for your letter and for the reviewers’ comments concerning our manuscript entitled “Fault diagnosis of a helical gearbox based on an adaptive empirical wavelet transform in combination with a spectral subtraction method” (ID: 456753). Those comments are all valuable and very helpful for revising and improving our paper, as well as the important guiding significance to our researches. We have studied comments carefully and have made the correction which we hope to meet with approval. Revised and deleted portion are marked in red and blue in the paper, respectively. The main corrections in the paper and the responds to the reviewer’s comments are as following:
Responds to the reviewer’s comments:
Point 1: Though the revised manuscript is well organised, the quality of accompanying letter is not maintain the standard. Authors should describe the detail of changes in accompanying letter mentioning section and page number. The experimental results are still insufficient which lacks the quality of the manuscript.
Response 1: Your comments are very important for us to improve the quality manuscript. Therefore, we add the comparative result of the frequency spectrum to support our idea by using spectral subtraction method. (p.8, lines 3-10). The experimental signal is used to test the effectiveness of the proposed algorithm. This paper focuses on discussing the advantages of the proposed algorithm in different conditions. In the future, we will use more experiment to study the optimization of the proposed algorithm based on your suggestion.
Furthermore, thank you for your suggestions. I’m sorry for inconvenience about the quality of accompanying letter, so a brief overview of the revision in the current manuscript is as following:
We add a link word between the first and second sentence (p.1, lines 13-14). Figure 1 and 2 are revised (p.4, line 1) and (p.10, lines 4), respectively. The manuscript is re-organized. The experiment analysis is in p.7, lines 26-33. The results are in p.9, lines 1-13. The dynamical model is established in p.9, lines 14-20 and p.10, lines 7-14. Then we add some simulated verification in different breakages (p.11, lines 4-21 and p.12, lines 2-9). Finally, the proposed algorithm has obvious advantages over other existed algorithms in the strong noise conditions (p.16, lines 2-15).
According to your suggestion, we have checked our manuscript by a native English speaking colleague.
Special thanks to you for your good comments.
We tried our best to improve the manuscript and made some changes in the manuscript. Here we did not list the changes but marked in red in revised paper.
We appreciate for Editors/Reviewers’ warm work earnestly and hope that the correction will meet with approval.
Once again, thank you very much for your comments and suggestions.
Reviewer 2 Report
I would like to thank the authors for improving the paper according my guidelines. As a hint for a further improvement of the paper, I would like to focus authors' attention to virtual sensors approach, which can potentially be applied to the gearbox systems
Virtual Diagnostic Sensors Design for an Automated Guided Vehicle Applied Sciences .- 2018, Vol. 8, no. 5, s. 1--20
Author Response
Dear Reviewers:
Thank you for your letter and for the reviewers’ comments concerning our manuscript entitled “Fault diagnosis of a helical gearbox based on an adaptive empirical wavelet transform in combination with a spectral subtraction method” (ID: 456753). Those comments are all valuable and very helpful for revising and improving our paper, as well as the important guiding significance to our researches. We have studied comments carefully and have made the correction which we hope to meet with approval. Revised and deleted portion are marked in red and blue in the paper, respectively. The main corrections in the paper and the responds to the reviewer’s comments are as following:
Responds to the reviewer’s comments:
Point 1: I would like to thank the authors for improving the paper according my guidelines. As a hint for a further improvement of the paper, I would like to focus authors' attention to virtual sensors approach, which can potentially be applied to the gearbox systems
Virtual Diagnostic Sensors Design for an Automated Guided Vehicle Applied Sciences .- 2018, Vol. 8, no. 5, s. 1--20.
Response 1: Thank you for your suggestions. We have read this paper in detail. For the dynamical model of our gearbox system, the vibration signal is measured from the radial acceleration of the shaft with ADAMS software. We consider your suggestion to study the virtual sensors for the fault diagnosis of gearbox system in the future.
According to your suggestion, we have checked our manuscript by a native English speaking colleague.
Special thanks to you for your good comments.
We tried our best to improve the manuscript and made some changes in the manuscript. Here we did not list the changes but marked in red in revised paper.
We appreciate for Editors/Reviewers’ warm work earnestly and hope that the correction will meet with approval.
Once again, thank you very much for your comments and suggestions.